# Comparative Study on Extracts from Traditional Medicinal Plants *Echinacea purpurea* (L.) Moench and *Onopordum acanthium* (L.): Antioxidant Activity In Vitro and Anxiolytic Effect In Vivo

**DOI:** 10.3390/ph18121801

**Published:** 2025-11-26

**Authors:** Maria Vlasheva, Mariana Katsarova, Ilin Kandilarov, Hristina Zlatanova-Tenisheva, Petya Gardjeva, Petko Denev, Kiril Atliev, Nora Sadakova, Maria Dimitrova, Ilia Kostadinov, Stela Dimitrova

**Affiliations:** 1Department of Bioorganic Chemistry, Faculty of Pharmacy, Medical University of Plovdiv, 15A Vassil Aprilov Blvd., 4002 Plovdiv, Bulgaria; mariya.vlasheva@mu-plovdiv.bg; 2Research Institute, Medical University of Plovdiv, 15A Vassil Aprilov Blvd., 4002 Plovdiv, Bulgaria; iliya.kostadinov@mu-plovdiv.bg; 3Department of Pharmacology and Clinical Pharmacology, Faculty of Medicine, Medical University of Plovdiv, 15A Vassil Aprilov Blvd., 4002 Plovdiv, Bulgaria; ilin.kandilarov@mu-plovdiv.bg (I.K.); hristina.zlatanova@mu-plovdiv.bg (H.Z.-T.); 4Department of Microbiology and Immunology, Faculty of Medicine, Medical University of Plovdiv, 15A Vassil Aprilov Blvd., 4002 Plovdiv, Bulgaria; petya.gardjeva@mu-plovdiv.bg; 5Laboratory of Biologically Active Substances, Institute of Organic Chemistry with Centre of Phytochemistry, Bulgarian Academy of Sciences, 139 Ruski Blvd., 4000 Plovdiv, Bulgaria; petko.denev@orgchm.bas.bg; 6Department of Epidemiology and Disaster Medicine, Faculty of Public Health, Medical University of Plovdiv, 15A Vassil Aprilov Blvd., 4002 Plovdiv, Bulgaria; kiril.atliev@mu-plovdiv.bg; 7Clinic of Neurology, St. Panteleimon Hospital Plovdiv, 9 Nicola Vaptsarov Blvd., 4004 Plovdiv, Bulgaria; norasadakova@abv.bg; 8Faculty of Pharmacy, Medical University of Plovdiv, 15A Vassil Aprilov Blvd., 4002 Plovdiv, Bulgaria; mimidimitr@gmail.com

**Keywords:** anxiety-like behavior, oxidative stress, interferon-gamma, interleukin-10, tumor necrosis factor-alpha, elevated plus maze, acute stress

## Abstract

**Background:** *Echinacea purpurea* (L.) Moench (EP) and *Onopordum acanthium* (L.) (OA) are promising medicinal plants with diverse biological activities but there is no information on the effects of their combinations. To harness the therapeutic potential of both while minimizing the risk of adverse effects, we prepared two combinations (CE1 and CE2) of EP and OA in ratios 1:1 and 3:1, respectively. **Methods:** Oxygen radical absorbance capacity (ORAC), hydroxyl radical absorbance capacity (HORAC), and an electrochemical assay were used to determine the antioxidant activity of the extracts in vitro. The anxiolytic and immunomodulatory properties were studied in rats. Animals were subjected to acute cold stress and anxiety-like behavior was evaluated by the elevated plus maze (EPM) and social interaction test (SIT). Serum IFN-γ, TNF-α and IL-10 levels were measured by ELISA. **Results:** CE2 demonstrated the highest antioxidant activity (1841.7 μmolTE/g by ORAC, 277.2 GAE/g by HORAC, and 39.6 by electrochemical method). Moreover CE2 produced anxiolytic-like effects—significantly increasing the open arms entries ratio (OAER; *p* < 0.001), open arms time ratio (OATR; *p* < 0.01) in the EPM, and prolonging the social interaction time (*p* < 0.05) versus the stressed control. OA increased OAER (*p* < 0.01) and OATR (*p* < 0.001), while EP increased only OAER (*p* < 0.01). CE1 showed no significant behavioral consequences. CE2 significantly reduced IFN-γ (*p* < 0.05), and IL-10 levels were elevated in OA and CE2 groups (*p* < 0.01). No significant changes in TNF-α levels were observed across groups. **Conclusions:** These findings indicate that CE2 and OA attenuate anxiety-like behavior and modulate the immune response primarily by stimulating IL-10 production.

## 1. Introduction

Anxiety- and stress-related disorders are increasingly understood as conditions in which dysregulated oxidative and inflammatory mechanisms play a key pathophysiological role. One of the primary causes of anxiety disorders is stress, which represents the body’s natural response to unexpected or challenging situations. Stress initiates a complex physiological reaction characterized by increased heart rate, blood pressure, and blood glucose levels, triggering the classical “fight or flight” response [1]. In addition to such acute reactions, persistent exposure to stressful events leads to chronic stress, which, over time, results in systemic deterioration. Chronic stress disrupts homeostasis, producing sustained anxiety and fear that may progress to depression [2]. Several hypotheses are offered to explain these effects, including the theory of inflammatory, oxidative, and nitrosative stress [3]. A substantial body of preclinical and clinical evidence supports the association between anxiety, depressive disorders, and oxidative stress [4,5,6].

Oxidative stress is linked to increased levels of pro-inflammatory cytokines, highlighting the tight interconnection between redox and immune processes [7]. In this context, T helper type 1 (Th1)–associated cytokines (e.g., IL-1β, IL-2, IFN-γ, TNF-α) largely drive pro-inflammatory responses, whereas T helper type 2 (Th2)–associated cytokines (e.g., IL-10) exert predominantly anti-inflammatory and regulatory effects; a balanced Th1/Th2 profile is therefore critical for immune homeostasis [8]. Persistent distorting of this balance toward a pro-inflammatory state contributes to chronic low-grade inflammation, neuronal damage, and behavioral alterations relevant to depression and anxiety disorders [8,9]. Pro-inflammatory cytokines stimulate free-radical generation to counteract pathogens, while oxidative stress further amplifies inflammatory signaling and cytokine release, creating a cascade in which excessive free-radical production can trigger and maintain inflammatory cascades [10].

Endogenous antioxidant defenses are not always sufficient to counteract these effects, requiring additional antioxidant supplementation. Considerable research has therefore focused on identifying safe, naturally derived antioxidants [6,11,12]. These include plant extracts and secondary metabolites such as alkaloids, terpenes, flavonoids, phenolic acids, saponins, lignans, and anthocyanins, which possess documented antioxidant [13,14,15] and anxiolytic activities [11,16,17]. Our most recent studies have focused on individual and combined hydroethanolic extracts of *Echinacea purpurea* (L.) (EP) and *Onopordum acanthium* (L.) (OA), both of which are rich in bioactive polyphenols.

EP contains high levels of chicoric and caftaric acids and is well recognized for its immunomodulatory, anti-inflammatory, and antioxidant properties [18]. Extracts from *Echinacea* spp. have been shown to alleviate anxiety in humans [19]. In animal studies, these extracts reduce anxiety-like behavior in rodents in paradigms such as the elevated plus maze, social interaction test, and social avoidance test, without markedly affecting locomotor activity, memory, or reward behaviors, suggesting a relative selectivity for anxiety-related endpoints [20,21]. These behavioral effects are thought to be mediated, at least in part, by EP alkylamides that interact with the endocannabinoid system via CB receptors, providing a mechanistic link between its immunomodulatory, antioxidant, and anxiolytic actions [22]. Furthermore, combinations of herbal extracts that include EP have been reported to prevent stress-induced social avoidance in mice, reduce brain expression of pro-inflammatory cytokines (TNF-α, IL-1β, and IL-6), and increase anti-inflammatory IL-10 and brain-derived neurotrophic factor (BDNF)/TrkB signaling [23].

OA, commonly known as Scotch thistle, contains significant amounts of myricetin and chlorogenic acid and has traditionally been used for its hepatoprotective and anti-inflammatory effects [24]. Experimental studies with methanolic OA leaf extracts have demonstrated potent antioxidant properties, attenuation of myocardial inflammation, and protection against β-cell injury in diabetic rats, indicating that OA can modulate oxidative and inflammatory processes relevant to stress-related pathology [25].

Plant-derived bioactive compounds have shown considerable potential in modulating immune function, mitigating inflammatory processes, and exerting neuroprotective effects [26]. Clinical studies support the efficacy of phytomedicines as adjunct or alternative treatments for anxiety, with several plant-based preparations showing good tolerability and greater efficacy than placebo [27]. Taken together, EP and OA exhibit complementary polyphenolic profiles and converge on shared antioxidant and immune-modulating pathways, providing a scientific rationale for comparing single extracts with defined EP–OA combinations in anxiety-related models.

The elevated plus maze (EPM) is a widely used, well-validated test of anxiety-like behavior in rodents, in which increased open-arm exploration reflects reduced anxiety and the paradigm is sensitive to both anxiolytic and anxiogenic agents [28,29]. The social interaction test (SIT) complements the EPM by capturing social aspects of anxiety, with reduced interaction between conspecifics indicating higher anxiety levels [30]. Using both tests in parallel allows complementary assessment of exploratory conflict and social behavior, supporting their selection for the present study.

Acute cold exposure represents a brief, controllable stressor that activates hypothalamic–pituitary–adrenal, autonomic, and immune responses, providing a valid experimental model for studying anxiety-like behavior in rodents [31,32]. When combined with validated behavioral tests such as the EPM or SIT, it reveals stress-sensitive behavioral alterations with predictive validity [31,33]. Compared with other stress models (e.g., restraint, social defeat, chronic unpredictable stress), cold exposure is non-painful and easily standardized. However, limitations include altered locomotion due to thermoregulatory drive, variability across species and sex, and inconsistent expression of classical anxiolytic patterns; therefore, findings should be interpreted as exhibiting “anxiolytic-like” effects [31,33,34]. Some protocols expose rats to 5 °C for 1–2 h/day in their home cages, increasing adrenal weight and plasma corticosterone levels and inducing anxiety-like behavior in males [33]. The behavioral outcomes depend on exposure temperature and duration, underscoring the need for precise reporting and cautious interpretation [34].

In previous work, we examined hydroethanolic extracts from EP and OA, as well as two defined combinations in ratios of 1:1 (CE1) and 3:1 (CE2) of EP to OA. The results showed significant immunomodulatory effects in rats with lipopolysaccharide-induced systemic inflammation, particularly pronounced in the combined formulations [35]. Given that immune dysregulation and oxidative stress contribute to the pathophysiology of anxiety disorders, these extracts and their combinations may possess anxiolytic-like properties. However, evidence regarding the anxiolytic potential of EP is limited, and, to our knowledge, no such data exist for OA. Furthermore, the biological effects of their combined formulations have not yet been investigated. The present study aims to address these gaps by evaluating the potential anxiolytic-like and immunomodulatory effects of EP, OA, and their combinations in an acute cold stress model, combining behavioral evaluation with serum analyses of IFN-γ, TNF-α, and IL-10. In addition, the antioxidant activity (AOA) of the extracts and their combinations was assessed in vitro using three established methods: ORAC, HORAC, and electrochemical assays [36,37,38]. This study extends our ongoing research on these extract combinations to further clarify their therapeutic potential and support their prospective clinical application.

## 2. Results and Discussion

### 2.1. Determination of AOA of Extracts of EP and OA and Their Combinations In Vitro

The results obtained in the present study are shown in Table 1. The AOA, as determined by the ORAC and HORAC assays, correlates directly with the polyphenol content of the extracts. The electrochemical method applies to antioxidants that exercise their effects through any of the three mechanisms—hydrogen atom transfer, electron transfer, and interaction with conjugated double bonds [39]. The combined use of these three assays provides a more comprehensive evaluation of the antioxidant potential of the obtained plant extracts.

Many studies have been devoted to assessing the antioxidant power of various substances or extracts, and for this purpose, various tests such as 2,2-diphenyl-1-picrylhydrazyl (DPPH), 2,2′-azino-bis(3-ethylbenzothiazoline-6-sulfonic acid) (ABTS), ferric reducing antioxidant power (FRAP), ORAC, peroxyl radical scavenging capacity (PSC), etc., have been applied [40]. However, there is still no universal method for measuring AOA, which makes it difficult to compare the results of different studies [41]. The results of the AOA of the medicinal plants studied by us, determined by the ORAC method, are shown in Table 1 (column 2). According to this method, the antioxidant activity of CE2 was the most pronounced, exceeding that of OA fivefold and that of EP 1.3-fold. The high concentration of chicoric and caftaric acids and rutin in the individual extract of EP and, respectively, of myricetin, quercetin, and arctigenin in that of OA (Table 2) contribute to the greatest extent to the activity of the combined extracts, taking into account the share of each of the plants in them. Several studies have confirmed the AOA of polyphenolic compounds [40], as well as their antidepressant effect [6,42]. Fu et al. found a high amount of chicoric and caftaric acids in the aerial parts of EP (42 mg/g and 17 mg/g, respectively) and reported high ORAC (1300 µmol TE/g) and ABTS-radical scavenging activity [43]. The AOA of extracts of OA with ethanol, water, methanol and sunflower oil was determined and values ranging from 1742.49 for the ethanol extract to 11.41 µmol TE/g for the oil extract were reported [44].

There are not many reports in the scientific literature on the use of the HORAC method for determining the AOA of extracts from EP or OA. Parzhanova et al. analyzed the same four extracts from OA by HORAC, reporting an activity of 716.37 GAE/g for the ethanolic one to 42.13 µmol GAE/g for the methanolic one, and no such activity for the oily ones [44]. In our experiments, we attained an activity of 60.0 µmol GAE/g for the extract from OA. By this method, the highest activity was shown by CE2—277.2 µmol GAE/g (Table 1, column 3) (being 4.6-fold higher than the AOA of OA and 1.5-fold higher than that of EP), which was not simply a mechanical sum of the activities of the individual extracts participating proportionally in it.

Electrochemical methods used to determine AOA are rapid, simple, sensitive, require a small amount of sample and have been successfully applied recently to analyze the antioxidant potential of polyphenols in foods [45,46,47]. An electrochemical method has been used to determine AOA of EP extracts and pharmaceutical forms [48,49] and is expressed in ascorbic or gallic acid equivalents per gram of extract. In our experiment, it was determined by a kinetic criterion that counts the amount of oxygen forms reacted with the sample over time and calculated relative to Trolox. The results are presented in Table 1 (columns 4, 5). The AOA of CE2 was three times higher than that of the OA extract and 1.8 times higher than that of the EP extract. Again, it is striking that the most pronounced activity by this method is exhibited by the CE1 and CE2, which is confirmation of the fact that the active substances act in combination [50]. Undoubtedly, plant extracts have AOA due to the content of polyphenolic compounds [51]. Having demonstrated the AOA of individual and combined extracts of EP and OA, it was interesting to establish whether it correlated with an anxiolytic effect in experimental animals subjected to acute cold stress and treated with the same extracts.

### 2.2. Study of the Anxiolytic Effect of EP and OA Extracts and Their Combinations in Experimental Animals Subjected to an Acute Cold Stress Model

#### 2.2.1. Confirmation of Stress Induction

An independent sample *t*-test was used to compare the two control groups in order to assess the effect of the stressor. Exposure to 5 °C for 1 h resulted in a significant decrease in social interaction time (*p* < 0.001), a reduced ratio of open arm entries to total entries (open arms entries ratio, OAER) (*p* = 0.003), and less time spent in the open arms of the EPM (*p* < 0.001). In contrast, time spent in the closed arms was increased (*p* < 0.001). Consequently, the ratio of time spent in open arms to total time (open arms time ratio, OATR) was significantly lower in the stressed control group (*p* < 0.001). These results indicated that acute cold exposure increases anxiety-like behavior in rats. No statistically significant differences were found in the number of entries into the closed arms of the maze or in the total number of arm entries between the non-stressed control and stressed control groups. The results are presented in Table 3.

#### 2.2.2. EPM

The number of closed arm entries and the total number of arm entries in the EPM were used as indicators of locomotor activity in the tested animals [52], as total distance traveled was not measured, which represents a limitation of the current study. Levene’s test confirmed that the assumption of homogeneity of variances was met (*p* = 0.31 and *p* = 0.07, respectively). However, one-way ANOVA revealed no statistically significant differences between the groups in either the number of closed arm entries (F_(4, 35)_ = 1.72, *p* = 0.17) or the total number of arm entries (F_(4, 35)_ = 1.86, *p* = 0.14) (Figure 1A and Figure 1B, respectively).

The OAER is considered a more reliable indicator of anxiolytic activity than the absolute number of open arm entries, as it helps minimize confounding effects related to general locomotor activity. Homogeneity of variances was met (Levene’s test: *p* = 0.073) and one-way ANOVA was used to assess differences between groups. The analysis revealed a statistically significant effect (F_(4, 35)_ = 6.834, *p* < 0.001). Tukey post hoc test showed that in rats treated with individual extracts, CE1 and CE2, this ratio was significantly increased compared to the stressed control group (*p* < 0.01, *p* < 0.05, and *p* < 0.001, respectively), suggesting a potential anxiolytic-like effect of these treatments. The results are presented in Figure 1C.

Time spent in the open arms of the maze is considered an indicator of anxiolytic-like effect, whereas time spent in the closed arms reflects increased anxiety levels. Since homogeneity of variances was not met (Levene’s test: *p* = 0.004), Welch’s ANOVA was used to evaluate differences in these times between groups. The analysis indicated a statistically significant effect (F_(4, 15.27)_ = 30.46; *p* < 0.001). Games–Howell post hoc test showed that rats treated with extracts OA and CE2 spent significantly more time in the open arms of the maze compared to the stressed control group (*p* < 0.001, and *p* < 0.01, respectively), and those treated with extracts EP (*p* < 0.001, and *p* < 0.05, respectively) and CE1 (*p* < 0.001, and *p* < 0.05, respectively) (Figure 1D). On the contrary, animals in the OA and CE2 groups spent less time in the closed arms of the maze (Figure 1E).

The OATR is another parameter used to assess anxiety-reducing activity in the EPM. The assumption of homogeneity of variances was violated (Levene’s test: *p* = 0.003). However, the Welch’s ANOVA indicated a significant effect (F_(4, 15.23)_ = 28.02; *p* < 0.001). Post hoc analysis using the Games–Howell test revealed that this ratio was significantly higher in rats treated with extracts OA and CE2 compared to the stressed control group (*p* < 0.001, and *p* < 0.01, respectively) and the groups treated with EP (*p* < 0.001, and *p* < 0.05, respectively) and CE1 (*p* < 0.001, and *p* < 0.05, respectively) (Figure 1F). These data supported the anxiolytic potential of OA and CE2 in the tested animals.

#### 2.2.3. SIT

Social interaction time was increased in rats treated with OA, CE1, and CE2 compared to those in the C-stress and EP groups. As Levene’s test showed that the assumption of homogeneity of variances was not met (*p* = 0.007), the Welch’s ANOVA was used. It appeared to be significant (F_(4, 17.01)_ = 6.80; *p* = 0.002) and Games–Howell post hoc was performed to compare differences between groups. It revealed that the increase in social interaction time was statistically significant in the CE2 group compared to both the stressed control and EP-treated groups (*p* < 0.05). Although groups OA and CE1 did not reach statistical significance, their values did not significantly differ from those of the CE2 group. These results are presented in Figure 2.

The anxiolytic-like effects of EP, OA, and their combinations were assessed using two behavioral paradigms: EPM and SIT. The EPM relies on the innate preference of rodents for dark, enclosed spaces. Increased ratios of OAER and OATR were considered indicative of anxiolytic-like effects, whereas the number of entries into closed arms and total arm entries were used as measures of general locomotor activity [52], although total distance traveled is generally regarded as a more accurate predictor of the latter. It was not measured in the current study due to technical limitations. Ratios provided a more accurate assessment of anxiety-related behavior than raw measures, such as absolute open arm entries or time spent in open arms, as they reduce the influence of overall locomotor activity. In the present study, extracts EP, OA, and CE2 exhibited anxiety-reducing effects, with OA and particularly CE2 showing more pronounced activity. In the SIT, significant anxiolytic-like effects were observed only in animals treated with CE2.

The observed anxiolytic-like effects may be attributed to the flavonoid content of the extracts, consistent with prior studies reporting antioxidant and antidepressant properties of rutin, quercetin, myricetin, and phenolic acids, including caffeic, ferulic, and chlorogenic acids [40,42]. Haller et al. demonstrated anxiolytic effects of certain *Echinacea* extracts, although only one extract was effective across a broad dose range [20]. Literature data on anxiety-reducing effect in animals and humans mainly concern another representative of the genus (*Echinacea angustifolia*) [21,53]. A recent study has demonstrated that in rats with bifenthrin-induced neurotoxicity, the hydro-ethanolic extract of EP reduced anxiety-like behavior and decreased the elevated levels of pro-inflammatory cytokines (TNF-α, IL-1β) as well as the apoptotic protein caspase-3 in the cortex and hippocampus [54]. However, no prior data exist on the anxiolytic activity of OA to our knowledge. In this study, OA exhibited anxiolytic potential in the EPM, likely due to its high content of chlorogenic acid and myricetin, which are absent in EP. Chlorogenic acid has established anxiolytic and antidepressant properties through neuroprotective, antiapoptotic, and anti-inflammatory mechanisms [55], and it has been reported to enhance γ-aminobutyric acid (GABA) neurotransmission [56]. Additionally, chlorogenic acid, like other phenolic acids, exhibits significant antioxidant activity [57].

### 2.3. Effects of EP and OA Extracts and Their Combinations on Serum Cytokine Levels

The effects of cold exposure on serum cytokine levels were assessed by comparing TNF-α, IFN-γ, and IL-10 between non-stressed and stressed control groups using an independent samples *t*-test. Acute cold stress resulted in a significant increase in IFN-γ (*p* = 0.026) and a significant decrease in IL-10 (*p* = 0.004) compared with non-stressed controls. Serum TNF-α showed an elevation that did not reach statistical significance (*p* = 0.14). These findings indicate that acute cold exposure induced a pro-inflammatory state characterized by increased IFN-γ and reduced IL-10 levels (Table 4).

#### 2.3.1. Changes in Serum TNF-α in Acute Cold Stress-Challenged Rats

Treatment with EP, OA, CE1, and CE2 reduced serum TNF-α levels compared with the stressed control group. The assumption of homogeneity of variances was violated (Levene’s test: *p* < 0.001). Welch’s ANOVA indicated that group differences were not statistically significant (F_(4, 16.67)_ = 2.62; *p* = 0.073). These results suggest a non-significant trend toward lower TNF-α levels following extract administration (Figure 3).

#### 2.3.2. Changes in Serum IFN-γ in Acute Cold Stress-Challenged Rats

All extracts reduced IFN-γ concentrations compared with the stressed control. The assumption of homogeneity of variances was not met (Levene’s test: *p* = 0.027), and Welch’s ANOVA showed a significant overall effect (F_(4, 16.38)_ = 3.70; *p* = 0.025). Games–Howell post hoc analysis revealed a significant decrease in serum IFN-γ only in the CE2-treated group compared with the stressed control (*p* < 0.05). Reductions in the other extract-treated groups did not reach statistical significance (Figure 4).

#### 2.3.3. Changes in Serum IL-10 in Acute Cold Stress-Challenged Rats

The assumption of homogeneity of variances was violated (Levene’s test: *p* = 0.006). Welch’s ANOVA indicated a significant overall effect (F_(4, 16.57)_ = 15.10; *p* < 0.001). Games–Howell post hoc analysis showed that OA and CE2 treatment significantly increased IL-10 concentrations compared with the stressed control (*p* < 0.01). CE2-treated animals also exhibited significantly higher IL-10 levels than those treated with EP (*p* < 0.05) and CE1 (*p* < 0.01). OA treatment resulted in higher IL-10 levels than CE1 (*p* < 0.01). These results indicate that OA and CE2 exerted the strongest stimulatory effects on IL-10 production (Figure 5).

### 2.4. Correlations Between Behavioral and Immunological Parameters of EP and OA Extracts and Their Combinations

Pearson correlation analysis was used to assess relationships between serum levels of IL-10 and INF-γ and behavioral parameters, including OAER, OATR, and social interaction time. A moderate positive correlation was observed between IL-10 concentrations and OATR (r(38) = 0.552; *p* < 0.001), suggesting that increased IL-10 may be associated with reduced anxiety-like behavior following extract administration. After Bonferroni correction for multiple testing (adjusted *p* < 0.008), the IL-10–OATR correlation remained statistically significant. Weak positive correlations between IL-10 and social interaction time, and weak negative correlations between IFN-γ and EPM ratios were found, though these did not reach statistical significance (Table 5). The absence of significant correlations for some parameters may reflect the limited sample size and inter-individual variability, which reduces statistical power. Further studies with larger sample sizes are warranted to clarify the potential mechanistic relationship between the anxiolytic and immunomodulatory effects of the extracts.

Serum IL-10 levels were plotted against the OATR for each treatment group to visualize the association between IL-10 and anxiety-like behavior. The scatter plot showed that data points from the CE2-treated group clustered toward higher IL-10 and OATR values, suggesting that IL-10 upregulation may contribute to the anxiety-reducing effects observed, particularly in CE2 (Figure 6).

Acute stress responses represent physiological defense mechanisms that can transiently stimulate inflammatory activity. While short-term immune modulation may be adaptive, persistent low-grade inflammation has damaging effects and contributes to various chronic diseases. Numerous studies have shown that stress responses involve elevated blood glucose, blood pressure, heart rate, and inflammatory mediators, including cytokines. Chronic stress has also been linked to conditions such as insulin resistance, cardiovascular disease, and cancer. Acute stress, however, exerts differential effects on various immune components, including circulating leukocyte counts, immune function parameters, and cytokine production [58].

Acute cold exposure increases metabolic activity and reactive oxygen species production while often diminishing the activity of antioxidant defense enzymes, creating an imbalance that contributes to oxidative stress [59]. The antioxidant activity of the extracts likely contributed to the observed anxiolytic-like effects, with CE2 demonstrating the highest activity and the most pronounced behavioral effects in both the EPM and SIT. Acute stress is also associated with immune activation and inflammation [60], and meta-analyses indicate that it increases serum levels of both pro- and anti-inflammatory cytokines [61]. Given the immunomodulatory properties of EP and OA, their potential contribution to anxiolytic-like effects was evaluated. Previous studies using immobilization stress showed that cold-pressed juice from EP aerial parts reduced serum IL-10, IL-6, and IL-17, as well as mRNA expression of these cytokines in the spleen [62]. In our study, control animals subjected to acute cold stress exhibited elevated IFN-γ levels (*p* = 0.026) and decreased IL-10 levels (*p* = 0.004) relative to unstressed controls, indicating a pro-inflammatory state induced by acute stress. Animal and human studies suggest that IFN-γ is associated with anxious and depressive behaviors [63,64], whereas IL-10 deficiency is linked to anxiety-like behaviors in mice [65] and reduced serum IL-10 levels are observed in patients with major depressive disorder [66]. Although TNF-α is constitutively expressed in the nervous system and its dysregulation is linked to anxiety-like behavior [67], its levels did not show significant changes following extract administration in our study. In the acute cold stress model, OA and CE2 markedly increased serum IL-10, and CE2 significantly reduced IFN-γ levels. These immunomodulatory effects corresponded with behavioral outcomes in the EPM, where these extracts elicited notable anxiolytic-like effects. Pearson correlation analyses revealed a significant positive correlation between serum IL-10 levels and OATR, suggesting that higher IL-10 concentrations were associated with reduced anxiety-like behavior. In contrast, correlations between IFN-γ and behavioral measures were weak and non-significant, indicating a lesser role for this pro-inflammatory cytokine. Collectively, these data suggest that modulation of the immune response, particularly IL-10 enhancement, may play a role in the anxiolytic potential of the extracts, especially CE2.

The immunomodulatory properties of the extracts are likely attributable to their flavonoid and phenolic acid content. Our previous analyses confirmed significant chlorogenic acid levels in both individual extracts and their combinations (Table 2). Shah et al. reported that chlorogenic acid suppressed microglial activation and reduced NF-κB and pro-inflammatory cytokines TNF-α and IL-1β in a rat model of middle cerebral artery occlusion [68]. Similarly, in a mouse model of 1-methyl-4-phenyl-1,2,3,6-tetrahydropyridine (MPTP)-induced neurotoxicity, chlorogenic acid suppressed NF-kB activation, reduced IL-1β and TNF-α expression, and upregulated IL-10 [69].

Myricetin, present in OA and combination extracts, inhibited TNF-α-induced NF-κB activation in cell culture [70] and increased IL-10 while reducing hippocampal TNF-α in a mouse model of neuroinflammation [71]. Other flavonoids and phenolic acids identified in OA and combined extracts, including arctigenin and caffeic acid, also demonstrated neuroprotective and immunomodulatory effects, including IL-10 upregulation [72,73,74].

## 3. Materials and Methods

### 3.1. Plant Material from EP and OA

Dry aerial parts of EP and flowers of OA (Asteraceae) were obtained from Herb Pharmacy 36.6, Plovdiv, Bulgaria, accompanied by quality certificates from MediHerb-83 Ltd., Plovdiv, Bulgaria. The dried plant material was powdered, and individual and combined extracts were prepared with drug-to-extract ratios of *E. purpurea* to *O. acanthium* were 1:1 (CE1) and 3:1 (CE2) [35].

### 3.2. Extraction and HPLC Analysis of Individual Compounds

The extractions were performed with 60% ethanol. Subsequently, the ethanol was completely evaporated under reduced pressure, yielding a concentrated plant extract with no residual ethanol content. The typical EP and OA phenolic compounds were determined according to the method previously described by Vlasheva et al. using an HPLC system ProStar 230 solvent delivery module and photodiode array detector model 335 (Varian, Belrose, Australia); Hitachi C18 AQ (250 mm × 4.6 mm, 5 μm) column (Hitachi, Tokyo, Japan) [35].

### 3.3. Determination of AOA In Vitro

#### 3.3.1. Oxygen Radical Absorbance Capacity (ORAC) Assay

The method developed by Ou et al. was used with some modifications [36]. This method measures the ability of an antioxidant to neutralize peroxyl radicals. It is based on the inhibition of the decline in fluorescence of fluorescein during its oxidation in the presence of an antioxidant. The thermal decomposition of 2,2′-azobis(2-amidinopropane) dihydrochloride (AAPH) is used as a peroxyl radical generator. Briefly, 170 µL of fluorescein (70 nmol/L) and 10 µL of sample were incubated for 20 min at 37 °C directly in the apparatus. Then, 20 µL of AAPH (51.5 mM final concentration) was added to the reaction mixture. The final reaction volume was 200 µL, and all solutions were prepared in phosphate buffer (75 mM, pH = 7.4). The mixture was shaken and the fluorescence was read every minute until zero value was reached. AOA was expressed by comparison with the results obtained for standard Trolox solutions with concentrations (3.125, 6.25, 12.5, 25 and 50 µM), based on which a standard curve was constructed. The concentration of antioxidants in the sample is directly proportional to the area under the fluorescence decay curve. One ORAC unit is taken to be the area under the fluorescence decay curve of a Trolox solution with a concentration of 1 μM. Results were expressed as μmol Trolox equivalents per gram of extract (μmol TE/g). Measurements were performed using a FLUOstar OPTIMA fluorometer (BMG LABTECH, Offenburg, Germany) at excitation/emission wavelengths of 485/520 nm. All assays were conducted in triplicate.

#### 3.3.2. Hydroxyl Radical Averting Capacity (HORAC) Assay

The method was developed by Ou et al. and measures the ability of an antioxidant to assess complex formation under Fenton reaction conditions, caused by the interaction between Co(II) and H_2_O_2_ [37]. Briefly, 170 μL of fluorescein (final concentration 60 nM) prepared in phosphate buffer (75 mM, pH = 7.4) and 10 μL of the sample were incubated at 37 °C for 20 min directly in the apparatus. Then, 10 μL of H_2_O_2_ (27.5 mM, final concentration) and 10 μL of cobalt solution (Co(II), 230 μM final concentration) were added to the reaction mixture. After shaking, the initial fluorescence was measured and measurements were made every minute until zero fluorescence was reached. Gallic acid solutions (31.25, 62.5, 125, 250 and 500 μM) prepared in phosphate buffer (75 mM, pH = 7.4) were used to construct a standard curve. One HORAC unit is taken to be the area under the fluorescence decay curve of a gallic acid solution with a concentration of 1 μM. The results are expressed in µmol gallic acid equivalents per gram of extract (μmol GAE/g). Measurements are performed on the FLUOstar OPTIMA fluorometer (BMG LAB-TECH, Offenburg, Germany). The excitation wavelength of 485 nm and emission wavelength of 520 nm were used. All assays were conducted in triplicate.

#### 3.3.3. Electrochemical Method for Determination of AOA

The electrochemical method was used to determine the AOA [38]. The experiment’s methodology consists of taking a voltamperogram of cathodic electroreduction of oxygen using the “Analyst AOA” (RU.C.31.113.A N28715, Tomsk, Russia), connected to a PC. AOA was calculated using the kinetic criterion (K, μmol/L·min), comparing sample values to the Trolox standard according to the equation:AOA = Ksample/Ktrolox(1)

Each sample was tested in triplicate (n = 3).

### 3.4. Experimental Animals and Treatment In Vivo

#### 3.4.1. Experimental Animals

Male Wistar rats with an average weight of 180 to 200 g were used. The animals were housed in cages (four animals per cage) and kept under standard laboratory conditions: a 12:12 h dark/light cycle, 45% relative humidity, a temperature of 26.5 °C ± 1 °C, and free access to food and water.

All experimental procedures were conducted following the Directive 2010/63/EU of the European Parliament and of the Council of 22 September 2010 on the protection of animals used for scientific purposes and Regulation № 20 of 1 November 2012 on the minimum requirements for the protection and humane treatment of laboratory animals and the requirements for the facilities for their use, breeding, and/or supply issued by the Bulgarian Ministry of Agriculture and Food. The experiments were approved by the Committee on Animal Ethics of the Bulgarian Agency for Food Safety permit № 299 from 15 April 2021 and the decision of the Ethical Committee at MU Plovdiv with protocol № 3 from 20 May 2021.

#### 3.4.2. Experimental Groups and Treatments

Forty-eight male Wistar rats were randomly divided into six groups (n = 8) by an independent researcher who was not involved in the subsequent behavioral and immunological assessments, thereby ensuring allocation blinding. The groups were as follows: C_0_—non-stressed control group, C—stress—stressed control group, EP—treated with *E. purpurea* extract in dose 500 mg/kg body weight (bw); OA—treated with *O. acanthium* extract in dose 500 mg/kg bw, CE1—treated with Combination 1 in dose 500 mg/kg bw, CE2—treated with Combination 2 in dose 500 mg/kg bw (Table 6).

The amount of concentrated extract (described in Section 3.2) that each rat should receive according to its weight was calculated so that it received a dose of 500 mg/kg body weight [75]. The extracts were administered by oral gavage for 14 days before rats were subjected to an acute cold stress model and throughout the experiment, including 1 h before behavioral assessments. Control groups received distilled water via oral gavage throughout the experiment to control for handling-related stress.

#### 3.4.3. Acute Cold Stress Model and Experimental Design

Cold exposure induces oxidative stress through increased production of reactive oxygen species, elevates stress hormone levels (adrenocorticotropic hormone and epinephrine), and promotes anxiety-like behavior in male rats [33,76]. Therefore, we employed an acute cold stress model to evaluate the anxiolytic-like effects of the extracts.

On days 15 and 16, the experimental animals (except the non-stressed control) were placed in a refrigerator at 5 °C for 1 h, followed by the respective behavioral tests. The rats were placed in their standard home cages (in groups as normally housed) inside the refrigerator under light conditions. During the procedure, animals remained active, and no signs of hypothermia, immobility, or health deterioration were observed. The well-being of the animals was closely monitored throughout and after the stress sessions, and no adverse outcomes were recorded.

Fifteen minutes after the last testing, blood samples were taken, and serum was separated for analysis of IFN-γ, IL-10, and TNF-α levels using the ELISA method.

The timeline of the experiment is presented in Figure 7.

#### 3.4.4. Behavioral Assessment of Anxiolytic Effects

##### Elevated Plus Maze (EPM)

EPM was used to evaluate anxiety-like behavior in rats. The apparatus consisted of two open arms and two closed arms (50 cm length × 10 cm width) and two closed arms (50 cm length × 10 cm width × 40 cm wall height), arranged opposite to each other in a plus configuration and elevated 50 cm above the floor [77]. The test was performed one hour after stress exposure (on day 15) as described by López-Crespo G.A. et al. [52]. It was conducted in a single session without a prior training session. Each rat was placed on the central platform of the maze facing an open arm and allowed to freely explore for a total of 10 min, with the first 5 min serving as habituation and the subsequent 5 min for behavioral observation. A trained observer, blinded to the experimental groups, recorded the number of entries into the open and closed arms, as well as the time spent in each arm. Observations were made while seated quietly approximately 1 m from the maze to minimize disturbance. An arm entry was defined as the animal placing all four paws into the respective arm. The following parameters were recorded: time spent in open arms, time spent in closed arms, number of entries into open arms, number of entries into closed arms, and total arm entries. The ratio of open arm entries to total entries, as well as the ratio of time spent in open arms to total time, was calculated and used as measures of anxiety-like behavior. One limitation of our study is the absence of data on total distance traveled, which is considered the most reliable indicator of locomotor activity. Instead, we used the number of closed arm entries and the total number of arm entries as indirect measures of locomotor activity [52].

The ratio of open arm entries to total entries (OAER) was calculated using the following formula:OAER = (Nop/Ntotal)(2)
where Nop and Ntotal denote open- and total-arm entries, respectively.

The ratio of time spent in open arms to total time (OATR) was calculated using the following formula:OATR = (Topen/Ttotal)(3)
where Top and Ttotal denote time spent in open arms and total time, respectively.

The maze was cleaned with 70% ethanol between trials to prevent olfactory cues from influencing behavior.

##### Social Interaction Test (SIT)

The social interaction test was performed one hour after stress exposure (on day 16). It is used to evaluate anxiety-like behavior, as reduced social interaction time is indicative of heightened anxiety levels [78]. Each trial involved a test subject and a randomly selected conspecific from a different cage within the same group, both placed in a plastic test arena (60 × 60 × 40 cm; length × width × height) for a duration of 5 min. Prior to testing, all animals were acclimated to the test arena through a 5 min habituation session on the same day.

To facilitate identification, the unfamiliar conspecific was marked with a non-toxic red marker on its tail. Body weight measurements were recorded before the experiment to ensure appropriate pairing between the test subject and the introduced conspecific. The total duration of active social behaviors (sniffing, following, grooming, crawling over/under) was recorded by a trained observer, blinded to the experimental groups. Reduced social engagement was interpreted as indicative of anxiety-like or socially avoidant behavior [79].

The arena was cleaned with 70% ethanol and dried before the next trial to eliminate olfactory cues and maintain experimental consistency.

##### Immunological Assays

Rat TNF-α (Cat. No. 865.000), IFN-γ (Cat. No. 865.010), and IL-10 (Cat. No. 670.070) ELISA kits (DiaClone, Besançon, France) were used. Blood was collected into pyrogen- and endotoxin-free tubes, centrifuged for 10 min, and serum aliquots stored at −70 °C. Serum cytokine levels were measured according to the manufacturer’s instructions. All ELISA assays were performed on coded serum samples. The experimenters conducting the analyses were blinded to the group allocations to minimize bias.

Rat sera were diluted 1:2 and assayed in duplicate. Standards, internal controls, and diluted serum samples (100 μL/well) were pipetted onto 96-well plates pre-coated with monoclonal antibodies specific for each cytokine. After incubation for 2 h at room temperature (20–25 °C), the plates were washed four times with PBS containing 0.05% Tween-20. Enzyme-conjugated secondary antibodies were then added and incubated for 1 h at room temperature to form cytokine–antibody–enzyme complexes. After a second washing step, TMB substrate was added and the plates were incubated in the dark for 15 min at room temperature. The enzymatic reaction was stopped with 2 N H_2_SO_4_, and color development, proportional to cytokine concentration, was measured spectrophotometrically at 450 nm with a reference wavelength of 620 nm using a TECAN ELISA reader (Zürich, Switzerland). Cytokine concentrations were determined from standard curves and expressed in pg/mL.

The detection limits and assay variability reported by the manufacturer were as follows:IFN-γ: sensitivity ~ 10 pg/mL; range 31.25–1000 pg/mL; intra-assay coefficient of variation (CV) < 5%, inter-assay CV < 10%IL-10: sensitivity 1.5 pg/mL; range 31.25–1000 pg/mL; intra-assay CV < 5%, inter-assay CV < 10%TNF-α: sensitivity 15 pg/mL; range 31.25–1000 pg/mL; intra-assay CV < 10%, inter-assay CV < 10%

### 3.5. Statistical Analysis

Statistical analysis was conducted using IBM SPSS 23.0 software. An independent samples *t*-test was applied to compare the two control groups to determine the stressor’s effect. Levene’s test was used to assess the assumption of homogeneity of variances. If this assumption was violated (Levene’s test, *p* < 0.05), Welch’s ANOVA was performed to evaluate differences between the stressed control and treated groups in behavioral and immunological assays, followed by the Games–Howell post hoc test. If the assumption of homogeneity was met, one-way ANOVA was used, followed by Tukey’s Honestly Significant Difference (Tukey HSD) post hoc test for pairwise comparisons. Pearson correlation analysis was used to examine the relationships between serum cytokine levels and behavioral parameters. A *p*-value of <0.05 was considered statistically significant.

## 4. Conclusions

The extracts CE2 and OA attenuated anxiety-like behavior in rats subjected to acute cold stress. These findings provide preliminary evidence that such combinations may modulate stress responses, potentially through mechanisms involving antioxidant and immunomodulatory activity. To our knowledge, this is the first study to demonstrate an anxiolytic-like effect of OA. CE2 exhibited the most pronounced antioxidant and anxiolytic-like effects, which were associated with decreased serum IFN-γ levels and increased IL-10 concentrations. The enhanced efficacy of CE2 likely reflects the combined contribution of bioactive constituents from both extracts. Overall, our results suggest that EP and OA, particularly in combination, may exert anxiolytic-like and immunomodulatory effects. Further investigations should explore dose–response relationships, chronic stress paradigms, and underlying molecular mechanisms to clarify their therapeutic potential in anxiety-related disorders.

This study has several limitations. Only a single dose was evaluated, and the findings are based on one acute stress model. Additionally, locomotor activity was not directly quantified (e.g., total distance traveled in the EPM).

## Figures and Tables

**Figure 1 pharmaceuticals-18-01801-f001:**
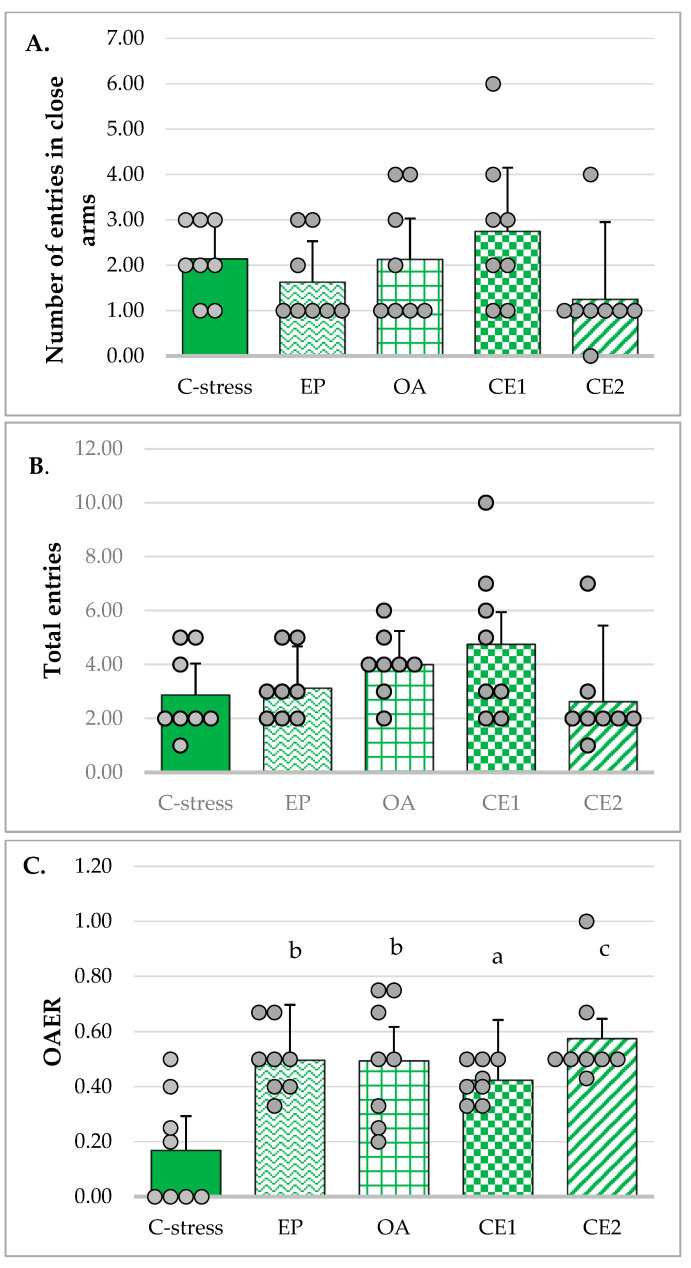
Effect of EP, OA, CE1, and CE2 on behavioral parameters in the EPM. Data are presented as mean ± SD with individual data points (n = 8 per group). (**A**) Number of entries in closed arms. (**B**) Total number of entries. (**C**) Ratio of open arms entries to the total number of entries (open arms entries ratio, OAER). Statistical differences were determined using one-way ANOVA, followed by Tukey HSD post hoc test. (**D**) Time spent in open arms. Statistical differences were determined using Welch’s ANOVA, followed by Games–Howell post hoc test. (**E**) Time spent in closed arms. Statistical differences were determined using Welch’s ANOVA, followed by Games–Howell post hoc test. (**F**) Ratio of time spent in open arms to total time (open arms time ratio (OATR). Statistical differences were determined using Welch’s ANOVA, followed by Games–Howell post hoc test. ^a^ *p* < 0.05 compared to the C-stress; ^b^ *p* < 0.01 compared to C-stress; ^c^ *p* < 0.001 compared to C-stress; ^d^ *p* < 0.05 compared to CE1; ^f^ *p* < 0.001 compared to CE1: ^g^ *p* < 0.05 compared to EP; ^h^ *p* < 0.001 compared to EP.

**Figure 2 pharmaceuticals-18-01801-f002:**
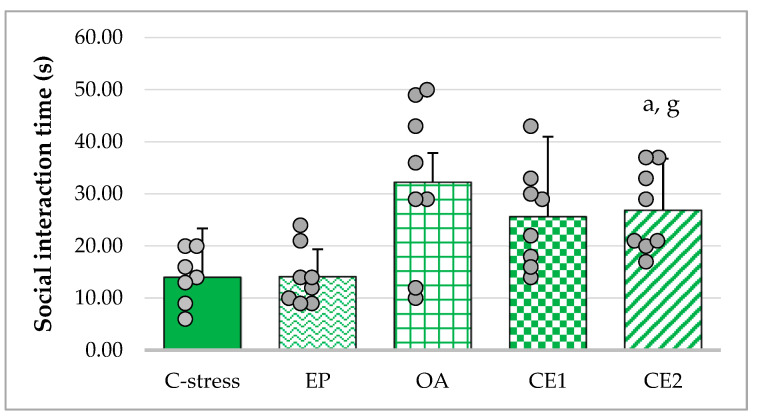
Effect of EP, OA, CE1, and CE2 on social interaction time. Data are presented as mean ± SD with individual data points (n = 8 per group). Statistical differences were determined using Welch’s ANOVA, followed by Games–Howell post hoc test. ^a^ *p* < 0.05 compared to C-stress; ^g^ *p* < 0.05 compared to EP.

**Figure 3 pharmaceuticals-18-01801-f003:**
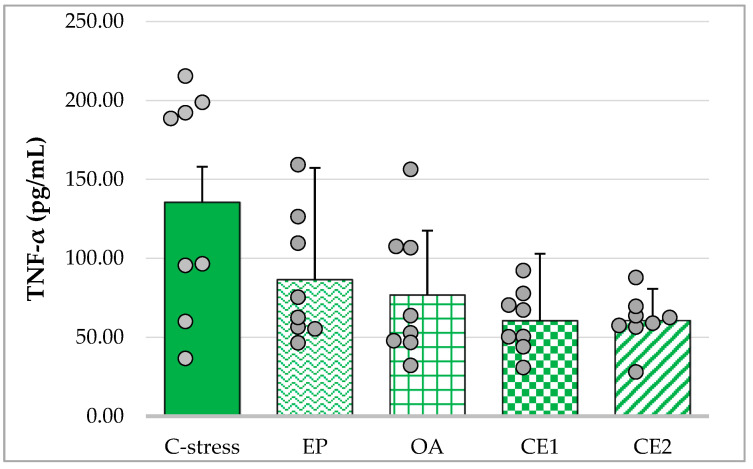
Effect of EP, OA, CE1, and CE2 on serum levels of TNF-α in rats subjected to acute cold stress. Data are presented as mean ± SD with individual data points (n = 8 per group).

**Figure 4 pharmaceuticals-18-01801-f004:**
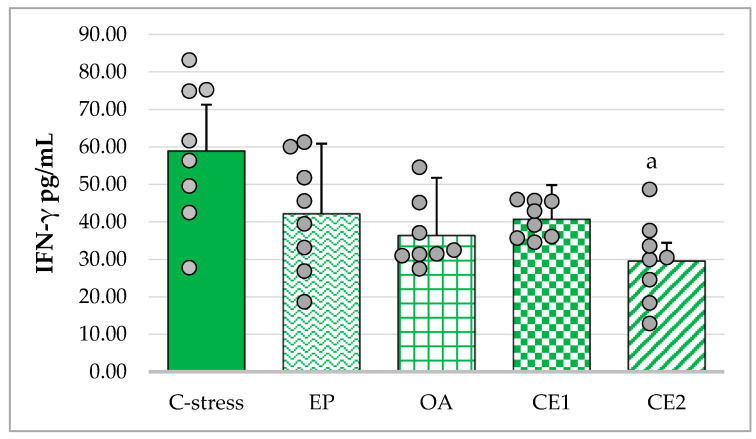
Effect of EP, OA, CE1, and CE2 on serum levels of IFN-γ in rats subjected to acute cold stress. Data are presented as mean ± SD with individual data points (n = 8 per group). Statistical differences were determined using Welch’s ANOVA, followed by Games–Howell post hoc test. ^a^ *p* < 0.05 compared to C-stress.

**Figure 5 pharmaceuticals-18-01801-f005:**
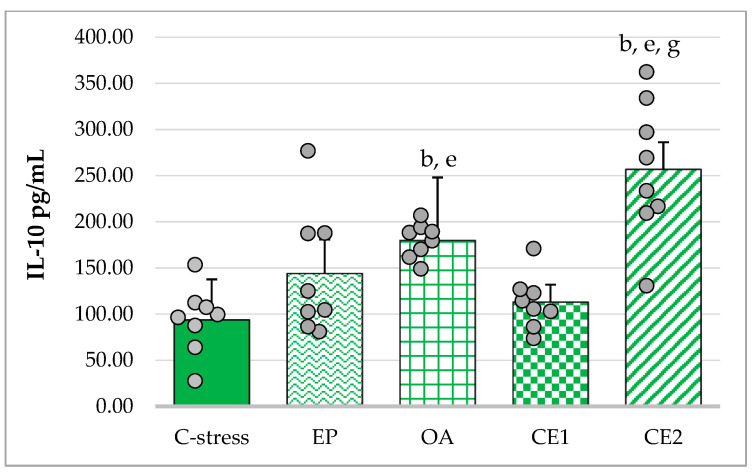
Effect of EP, OA, CE1, and CE2 on serum levels of IL-10 in rats subjected to acute cold. Data are presented as mean ± SD with individual data points (n = 8 per group). Statistical differences were determined using Welch ANOVA, followed by Games–Howell post hoc test. ^b^ *p* < 0.01 compared to C-stress; ^e^ *p* < 0.01 compared to CE1: ^g^ *p* < 0.05 compared to EP.

**Figure 6 pharmaceuticals-18-01801-f006:**
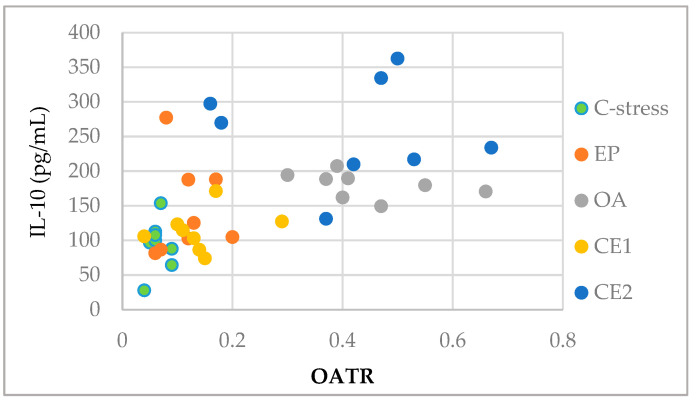
Scatter plot visualizing the association between serum IL-10 levels and OATR in rats across different treatment groups. Data points cluster by treatment, with the CE2 group showing generally higher IL-10 and OATR values.

**Figure 7 pharmaceuticals-18-01801-f007:**
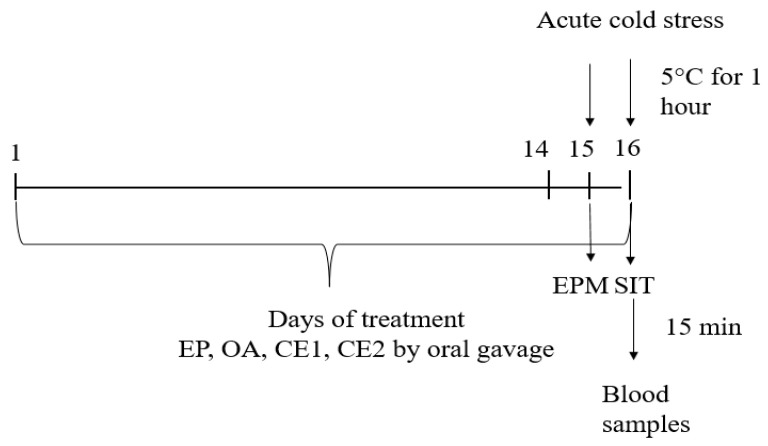
Schematic timeline of the experimental procedure. Extracts were administered at 500 mg/kg bw daily. Behavioral tests (EPM and SIT) and blood collection occurred at the indicated time points.

**Table 1 pharmaceuticals-18-01801-t001:** Antioxidant activity of extracts of *E. purpurea*, *O. acanthium*, and their combinations, measured by ORAC, HORAC, and electrochemical methods.

Extracts	ORAC, µmol TE/g	HORAC, µmol GAE/g	Electrochemical Method	
			K, μmol/L·min ± SD	AOA
EP	1417.2 ± 6.2	181.1 ± 10.4	14.0 ± 0.8	22.0
OA	368.4 ± 0.5	60.0 ± 1.0	8.5 ± 0.04	13.4
CE1	736.5 ± 6.3	174.4 ± 1.7	17.1 ± 0.9	27.0
CE2	1841.7 ± 75.0	277.2 ± 4.2	25.1 ± 1.3	39.6
Trolox	-	-	0.6 ± 0.001	1.0

Results are presented as mean value ± SD (n = 3); TE—trolox equivalent; GAE—gallic acid equivalent; K—kinetic criterion.

**Table 2 pharmaceuticals-18-01801-t002:** Content of biologically active substances in extracts of *E. purpurea*, *O. acanthium*, CE1, and CE2 (Reprinted from Ref. [35]).

	Extracts	EP	OA	CE1	CE2
Analyte, μg/g					
Ferulic acid		770.7 ± 44.9	nd	471.9 ± 28.26	726.0 ± 43.56
Caffeic acid		1115.0 ± 55.0	265.0 ± 14.1	696.0 ± 42.9	839.0 ± 24.9
Caftaric acid		3060.0 ± 142.3	nd	1450.0 ± 74.9	2748.0 ± 147.5
Chicoric acid		12,915.7 ± 773.2	nd	6505.0 ± 390.3	8350.0 ± 441.2
Cynarin		39.3 ± 2.1	nd	nd	trace
Echinacoside		55.4 ± 2.3	nd	nd	trace
Neochlorogenic acid		301.0 ± 27.3	596.0 ± 35.3	662.7 ± 49.8	443.8 ± 24.5
Chlorogenic acid		904.7 ± 54.1	661.0 ± 37.3	967.0 ± 55.5	330.6 ± 17.5
Quercetin		270.0 ± 13.3	584.6 ± 33.3	338.0 ± 3.3	98.5 ± 5.1
Apigenin		nd	280.0 ± 3.7	173.2 ± 2.7	57.5 ± 2.9
Rutin		2300.0 ± 132.1	nd	1340.0 ± 76.5	1837.0 ± 115.3
Myricetin		nd	1322.0 ± 66.3	1006.0 ± 65.7	361.0 ± 24.7
Epicatechin		142.3 ± 6.7	139.0 ± 1.2	856.0 ± 51.3	239.0 ± 15.3
Scutellarin		nd	35.0 ± 1.3	trace	nd
Arctigenin		nd	555 ± 32.7	225.2 ± 12.3	108.0 ± 13.2

Results are presented as mean value ± SD (n = 3); nd, undetected.

**Table 3 pharmaceuticals-18-01801-t003:** Comparison between stressed and non-stressed control in behavioral tests (EPM and SIT).

Parameter	Non-Stressed Control (n = 8) Mean ± SD	Stressed Control (n = 8) Mean ± SD	t(df)	*p*-Value
Total entries	3.75 ± 1.16	2.88 ± 1.55	t(14) = 1.28	*p* = 0.22 ^a^
Closed arms entries	2.00 ± 0.76	2.13 ± 0.83	t(14) = −0.31	*p* = 0.76 ^a^
OAER	0.47 ± 0.12	0.17 ± 0.2	t(14) = 3.56	*p* = 0.003 **^,a^
Time spent in open arms (s)	54.38 ± 12.37	19.38 ± 5.53	t(9.69) = 7.30	*p* < 0.001 ***^,b^
Time spent in closed arms (s)	245.62 ± 12.37	280.62 ± 5.53	t(9.69) = −7.30	*p* < 0.001 ***^,b^
OATR	0.18 ± 0.04	0.07 ± 0.02	t(14) = 8.05	*p* < 0.001 ***^,a^
Social interaction time (s)	32.25 ± 8.17	14.00 ± 4.87	t(14) = 5.43	*p* < 0.001 ***^,a^

OAER = open arms entries ratio; OATR = open arms time ratio; Independent sample *t*-tests were used for all comparisons; data are presented as mean ± SD; ^a^ equal variances assumed based on Levene’s test (*p* > 0.05), Student’s *t*-test applied; ^b^ equal variances not assumed based on Levene’s test (*p* < 0.05), Welch’s test applied; ** significance at *p* < 0.01; *** significance at *p* < 0.001.

**Table 4 pharmaceuticals-18-01801-t004:** Comparison of serum cytokine concentrations (pg/mL) between stressed and non-stressed controls.

Cytokine	Non-Stressed Control (mean ± SD)	Stressed Control (mean ± SD)	t(df)	*p*-Value
TNF-α (pg/mL)	93.23 ± 22.46	135.55 ± 70.75	t(8.40) = 1.61	*p* = 0.14 ^b^
IFN-γ (pg/mL)	39.23 ± 12.33	58.93 ± 18.72	t(14) = 2.49	*p* = 0.026 *^,a^
IL-10 (pg/mL)	164.34 ± 43.96	93.74 ± 36.71	t(14)= −3.49	*p* = 0.004 **^,a^

Independent samples *t*-tests were used for all comparisons; data are presented as mean ± SD; ^a^ equal variances assumed based on Levene’s test (*p* > 0.05), Student’s *t*-test applied; ^b^ equal variances not assumed based on Levene’s test (*p* < 0.05), Welch’s test applied; * significance at *p* < 0.05; ** significance at *p* < 0.01.

**Table 5 pharmaceuticals-18-01801-t005:** Correlation matrix between behavioral and cytokine parameters in rats exposed to acute cold stress.

Variables	INF-γ (pg/mL)	IL-10 (pg/mL)
OAER	r(38) = −0.103 *p* = 0.529	r(38) = −0.002 *p* = 0.999
OATR	r(38) = −0.299 *p* = 0.061	r(38) = 0.552 *p* < 0.001 ***
Social interaction time (s)	r(38) = −0.094 *p* = 0.563	r(38) = 0.255; *p* = 0.113

OAER = open arms entries ratio; OATR = open arms time ratio; Pearson’s correlation coefficients (r) are shown; *** significance at *p* < 0.001.

**Table 6 pharmaceuticals-18-01801-t006:** Distribution of experimental animals by group.

Group	Legend	Description	Acute Cold Stress Model
1	C-stress	Distilled water, 10 mL/kg body weight	Yes
2	C_0_	Distilled water, 10 mL/kg body weight	No
3	EP	*E. purpurea*, 500 mg/kg body weight	Yes
4	OA	*O. acanthium*, 500 mg/kg body weight	Yes
5	CE1	Combination 1, 500 mg/kg body weight	Yes
6	CE2	Combination 2, 500 mg/kg body weight	Yes

n = 8 per group.

## Data Availability

Data presented in this study is contained within the article.

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
