# Peer review of "Comparative Study on Extracts from Traditional Medicinal Plants *Echinacea purpurea* (L.) Moench and *Onopordum acanthium* (L.): Antioxidant Activity In Vitro and Anxiolytic Effect In Vivo"

_pharmaceuticals, 2025, doi:10.3390/ph18121801_

Round 1

Reviewer 1 Report

Comments and Suggestions for Authors

General comment

The manuscript by Vlasheva et al. presents a well-designed study exploring the antioxidant, behavioral, and immunomodulatory effects of Echinacea purpurea and Onopordum acanthium extracts, both individually and in combination, under an acute stress paradigm in rats. The experimental design is coherent and the methodological rigor—particularly in the blinding and statistical approach—is interesting. The integration of biochemical, behavioral, and cytokine analyses provides a comprehensive view of the extracts’ biological activity. Overall, it represents a valuable contribution to the field of phytotherapy and stress-related neuroimmunology.

Introduction

The introduction is comprehensive, scientifically sound, and provides a well-structured rationale linking oxidative stress, inflammation, and anxiety-related behavior. The integration of phytochemical background with behavioral paradigms is appropriate for the journal’s scope. However, the section would benefit from improved focus, and clearer transitions between biochemical mechanisms and behavioral hypotheses. Some sentences could be condensed to enhance flow and readability. Since the authors explore natural compounds, the citation of some actual references is pertinent:

Amarakoon D, Lee WJ, Tamia G, Lee SH. Indole-3-Carbinol: Occurrence, Health-Beneficial Properties, and Cellular/Molecular Mechanisms. Annu Rev Food Sci Technol. 2023, 14:347-366. doi: 10.1146/annurev-food-060721-025531.

Freire MAM, Natural compounds in neuroprotection and nervous system regeneration: a narrative review. Regen Med Rep. 2026, 10.4103, doi: 10.4103/REGENMED.REGENMED-D-25-00038.

Specific and structural comments

Lines 1–13: The introduction starts directly with a description of stress responses. Consider adding a brief framing sentence before that, stating the study’s general focus (e.g., “Anxiety and stress-related disorders are increasingly linked to oxidative and inflammatory mechanisms…”). This would anchor the reader early.

Lines 14–37: The Th1/Th2 cytokine paragraph is informative but dense. The authors could condense it by summarizing rather than enumerating each cytokine, focusing instead on the concept of immune balance disruption as a mechanistic link to anxiety.

Lines 58–86: The phytochemical characterization is strong, but citations could be unified to avoid excessive fragmentation. The section would read better if each plant were discussed in a single, cohesive paragraph, rather than alternating between compounds and effects. Suggestion: briefly mention why comparing EP and OA is scientifically relevant (e.g., complementary polyphenolic profiles, shared pathways in immune modulation). This helps justify the comparative design.

Lines 87–133: The EPM and SIT descriptions are well articulated but overly detailed for an introduction. Since these tests are standard, a concise summary is enough. As a suggestion, the authors could reduce the procedural details (e.g., “sniffing, following, grooming”) and keep focus on why both were chosen. This improves readability and avoids textbook-style exposition.

Minor suggestions:

  • Adopt in vivo and in vitro in italics throughout.
  • Maintain consistent tense: use present tense for general knowledge and past tense for specific previous findings.
  • Change “facilitating the ‘fight or flight’ response” → “triggering the classical ‘fight-or-flight’ response” for fluency.
  • Replace “the extracts and combinations’ antioxidant activity” → “the antioxidant activity of the extracts and their combinations” for clarity.

Methods

The Methods section is comprehensive and logically organized. It adheres to ethical and methodological standards and provides sufficient detail for replication. Nonetheless, several subsections contain redundant phrasing, inconsistent terminology (e.g., “extracts,” “combinations,” “formulations”), and occasional grammatical or stylistic issues that could be refined for conciseness and clarity. Standardizing units, spacing, and capitalization will also improve readability.

Specific Comments

3.1. Plant Material

  1. Please clarify the botanical identification procedure — indicate if plant material authentication was conducted by a botanist or relied solely on the supplier’s certificate
  2. Replace “drug ratio EP to OA in a 1:1 (CE1) and EP to OA in a 3:1 (CE2)” →
    “drug-to-extract ratios of E. purpurea to O. acanthium were 1:1 (CE1) and 3:1 (CE2).”
  3. Consider merging the two first sentences for flow:
    “Dry aerial parts of purpurea and flowers of O. acanthium (Asteraceae) were obtained from Herb Pharmacy 36.6 (Plovdiv, Bulgaria), accompanied by quality certificates from MediHerb-83 Ltd.”
  4. Typo: “deter-mined” → “determined.”

3.3. Determination of Antioxidant Activity in vitro

3.3.1. ORAC assay

  1. Replace “peroxid radicals” → “peroxyl radicals.”
  2. Express results uniformly: “Results were expressed as μmol Trolox equivalents per gram of extract (μmol TE/g).”
  3. Correct English syntax: “Measurements were performed using a FLUOstar OPTIMA fluorometer (BMG LABTECH, Offenburg, Germany) at excitation/emission wavelengths of 485/520 nm.”
  4. “Three independent replicates” → “All assays were conducted in triplicate.”

3.3.2. HORAC assay

  1. Correct “FLUOstarhu” → “FLUOstar.”
  2. Replace “form complexes in conditions of Fenton reaction” → “assess complex formation under Fenton reaction conditions.”
  3. Ensure units consistency: “μmol gallic acid equivalents per gram of extract (μmol GAE/g).”

3.3.3. Electrochemical method

  1. Correct typographical issues (“per litre/minute” → “per liter per minute”).
  2. Clarify: “Antioxidant activity (AOA) was calculated using the kinetic criterion (K, μmol/L·min), comparing sample values to the Trolox standard according to the equation AOA = Ksample/KTrolox.”
  3. Prefer: “Each sample was tested in triplicate (n = 3).”

3.4.2. Experimental groups and treatments

  1. Table legend should include the sample size (n = 8 per group).
  2. Clarify abbreviations at first use (EP, OA, CE1, CE2).
  3. Consider rewriting “to mimic the handling stress experienced by the treated groups” → “to control for handling-related stress.”
  4. Replace “distinct color (red)” → “a non-toxic red marker.”
  5. Correct “Reduced social engagement was interpreted as an indicator of social withdrawal and anxiety-like behavior” → “Reduced social engagement was interpreted as indicative of anxiety-like or socially avoidant behavior.”
  6. “The arena was cleaned with 70% ethanol…” → add “and dried before the next trial.”

Minor suggestions

  • Use purpurea and O. acanthium in italics consistently.
  • Replace “bw” → “body weight (bw)” on first appearance.
  • Keep all instrument names in parentheses.
  • Double-check numbering — there is a duplicated “3.3.3” (electrochemical method appears twice). Renumber to maintain sequential order.
  • Ensure consistent unit spacing: use “μmol/L·min,” “pg/mL,” and “°C” without extra spaces.

Results/Discussion

General comments

  1. Overall structure: The section effectively integrates results with discussion, but could benefit from clearer segmentation between data presentation and interpretation. In several paragraphs, interpretation precedes quantitative data, which can confuse readers. Consider reorganizing to first present findings, followed by their implications.
  2. Conciseness: Some paragraphs reiterate the same conclusion (e.g., the synergistic effect of CE2) multiple times. Streamlining these instances would improve readability and focus.
  3. Interpretative depth: While biochemical rationale is strong (e.g., linking chlorogenic acid and IL-10), some claims could be further substantiated with quantitative or comparative language. For example: “CE2 demonstrated the highest activity and the most pronounced behavioral effects.” Could specify “approximately X-fold higher than EP or OA alone.”

Minor suggestions

  • Replace repetitive phrasing (“anxiolytic-like effects”) occasionally with synonyms such as “anxiety-reducing activity” or “anxiolytic potential” for stylistic variation.
  • Use “respectively” sparingly — in several instances it can be removed for smoother readability.
  • Verify consistency of abbreviations (e.g., CE2 vs. C-E2, OAER vs. OAEr).

Author Response

We thank the reviewer for the helpful and valuable comments and recommendations, which will help improve the quality of the manuscript. Below, we provide point-by-point responses to the comments. All changes are highlighted in red.

Comment: Since the authors explore natural compounds, the citation of some actual references is pertinent:

Amarakoon D, Lee WJ, Tamia G, Lee SH. Indole-3-Carbinol: Occurrence, Health-Beneficial Properties, and Cellular/Molecular Mechanisms. Annu Rev Food Sci Technol. 2023, 14:347-366. doi: 10.1146/annurev-food-060721-025531.

Freire MAM, Natural compounds in neuroprotection and nervous system regeneration: a narrative review. Regen Med Rep. 2026, 10.4103, doi: 10.4103/REGENMED.REGENMED-D-25-00038.

Response: Citations have been added in the Introduction section.

Comment: Lines 1–13: The introduction starts directly with a description of stress responses. Consider adding a brief framing sentence before that, stating the study’s general focus (e.g., “Anxiety and stress-related disorders are increasingly linked to oxidative and inflammatory mechanisms…”). This would anchor the reader early.

Response: Added sentence: Anxiety and stress-related disorders are increasingly understood as conditions in which dysregulated oxidative and inflammatory mechanisms play a key pathophysiological role.

Comment: Lines 14–37: The Th1/Th2 cytokine paragraph is informative but dense. The authors could condense it by summarizing rather than enumerating each cytokine, focusing instead on the concept of immune balance disruption as a mechanistic link to anxiety.

Response:

The paragraph has been revised as recommended.

Former: Oxidative stress has been linked to increased levels of pro-inflammatory cytokines, highlighting the interconnection between oxidative and inflammatory processes [7]. Cytokines such as interleukin-1β (IL-1β), interleukin-2 (IL-2), interleukin-12 (IL-12), inter-feron-gamma (IFN-γ), and tumor necrosis factor-alpha (TNF-α) are T helper type 1 (Th1) cytokines with pro-inflammatory properties. When inadequately regulated, their activity may lead to tissue damage, as observed in autoimmune diseases. Contrarywise, interleu-kins-4 (IL-4), -5 (IL-5), -13 (IL-13), and the anti-inflammatory interleukin-10 (IL-10) belong to the T helper type 2 (Th2) cytokine group. The immune response depends on a balanced Th1/Th2 ratio [8]. Pro-inflammatory cytokines promote free radical generation to counteract pathogens, while oxidative stress, in turn, enhances inflammatory signaling and cytokine release. Consequently, excessive free radical production often initiates inflammatory cascades [9].

New: Oxidative stress has been linked to increased levels of pro-inflammatory cytokines, highlighting the tight interconnection between redox and immune processes [7]. In this context, T helper type 1 (Th1)–associated cytokines (e.g., IL-1β, IL-2, IFN-γ, TNF-α) largely drive pro-inflammatory responses, whereas T helper type 2 (Th2)–associated cytokines (e.g., IL-10) exert predominantly anti-inflammatory and regulatory effects; a balanced Th1/Th2 profile is therefore critical for immune homeostasis [8]. Persistent distorting of this balance toward a pro-inflammatory state contributes to chronic low-grade inflammation, neuronal damage, and behavioral alterations relevant to depression and anxiety disorders [8, Freire]. Pro-inflammatory cytokines stimulate free-radical generation to counteract pathogens, while oxidative stress further amplifies inflammatory signaling and cytokine release, creating a cascade in which excessive free-radical production can trigger and maintain inflammatory cascades [9].

Comment: Lines 58–86: The phytochemical characterization is strong, but citations could be unified to avoid excessive fragmentation. The section would read better if each plant were discussed in a single, cohesive paragraph, rather than alternating between compounds and effects. Suggestion: briefly mention why comparing EP and OA is scientifically relevant (e.g., complementary polyphenolic profiles, shared pathways in immune modulation). This helps justify the comparative design.

Response:

The paragraph has been revised as recommended.

Former: EP contains high levels of chicoric and caftaric acids and is well-recognized for its immunomodulatory, anti-inflammatory, and antioxidant properties [17]. OA, commonly known as Scotch thistle, contains significant amounts of myricetin and chlorogenic acid and has traditionally been used for its hepatoprotective and anti-inflammatory effects [18]. The presence of these polyphenols in EP and OA supports their potential for antioxidant, neuroprotective and immunomodulatory properties.

Extracts from Echinacea have been shown to reduce anxiety-like behavior in rodents in paradigms such as the elevated plus maze, social interaction, and social avoidance tests [19]. Subsequent studies confirmed these findings and demonstrated that the extracts did not significantly affect locomotion, memory, or reward behaviors, indicating relative selectivity [20]. The active alkylamides in Echinacea interact with the endocannabinoid sys-tem through CB receptors, which may mediate its effects on immune function, oxidative stress, and behavior [21]. In the case of OA, methanolic leaf extracts positively impacted myocardial inflammation and β-cell injury in diabetic rats and exhibited potent antioxi-dant properties, suggesting a role in stress-related processes [22]. Furthermore, combina-tions of herbal extracts (including Echinacea) have prevented stress-induced social avoid-ance in mice, decreased brain expression of pro-inflammatory cytokines (TNF-α, IL-1β, and IL-6), and increased anti-inflammatory IL-10 and brain-derived neurotrophic factor (BDNF)/TrkB signaling [23]. Clinical studies also support the efficacy of phytomedicines as adjunct or alternative treatments for anxiety, with several plant-based preparations showing good tolerability and greater efficacy than placebo [24].

New: EP contains high levels of chicoric and caftaric acids and is well recognized for its immunomodulatory, anti-inflammatory, and antioxidant properties [17]. Extracts from Echinacea have been shown to reduce anxiety-like behavior in rodents in paradigms such as the elevated plus maze, social interaction, and social avoidance tests, without markedly affecting locomotion, memory, or reward behaviors, suggesting relative selectivity for anxiety-related endpoints [19,20]. These behavioral effects are thought to be mediated, at least in part, by EP alkylamides that interact with the endocannabinoid system via CB receptors, providing a mechanistic link between its immunomodulatory, antioxidant, and anxiolytic actions [21]. Furthermore, combinations of herbal extracts that include Echinacea have been reported to prevent stress-induced social avoidance in mice, reduce brain expression of pro-inflammatory cytokines (TNF-α, IL-1β, and IL-6), and increase anti-inflammatory IL-10 and brain-derived neurotrophic factor (BDNF)/TrkB signaling [23].

OA, commonly known as Scotch thistle, contains significant amounts of myricetin and chlorogenic acid and has traditionally been used for its hepatoprotective and anti-inflammatory effects [18]. Experimental studies with methanolic OA leaf extracts have demonstrated potent antioxidant properties, attenuation of myocardial inflammation, and protection against β-cell injury in diabetic rats, indicating that OA can modulate oxidative and inflammatory processes relevant to stress-related pathology [22].

Plant-derived bioactive compounds have shown considerable potential in modulating immune function, mitigating inflammatory processes, and exerting neuroprotective effects [Amarakoon D, 2026]. Clinical studies support the efficacy of phytomedicines as adjunct or alternative treatments for anxiety, with several plant-based preparations showing good tolerability and greater efficacy than placebo [24]. Taken together, EP and OA exhibit complementary polyphenolic profiles and converge on shared antioxidant and immune-modulating pathways, providing a scientific rationale for comparing single extracts with defined EP–OA combinations in anxiety-related models.

Comment: Lines 87–133: The EPM and SIT descriptions are well articulated but overly detailed for an introduction. Since these tests are standard, a concise summary is enough. As a suggestion, the authors could reduce the procedural details (e.g., “sniffing, following, grooming”) and keep focus on why both were chosen. This improves readability and avoids textbook-style exposition.

Response:

The paragraph has been revised as recommended.

Former: Among ethological paradigms, the elevated plus maze (EPM) is considered the gold standard for assessing anxiety-like behavior in rodents. Based on rodents’ aversion to open, elevated areas, it measures the conflict between exploration and avoidance. Reduced anxiety is indicated by increased time spent and entries into the open arms [25, 26]. The EPM is well validated in both rats and mice, offering simplicity, reproducibility, and sen-sitivity to anxiolytic and anxiogenic agents. The social interaction test (SIT) complements the EPM by evaluating social dimensions of anxiety. It quantifies direct social behaviors (sniffing, following, grooming) between two conspecifics in a novel environment, with shorter interaction times reflecting higher anxiety levels. The SIT is particularly suited for studying social anxiety disorders and has demonstrated sensitivity to anxiolytic and anxiogenic compounds, providing insights into neurobiological mechanisms not cap-tured by conflict-based models [27].

Together, the EPM and SIT offer complementary perspectives on anxiety – the former addressing exploratory conflict and the latter evaluating social behavior. Their combined use ensures strong face and predictive validity, along with practical and cost-effective im-plementation, justifying their selection for the present study.

New: The elevated plus maze (EPM) is a widely used, well-validated test of anxiety-like behavior in rodents, in which increased open-arm exploration reflects reduced anxiety and the paradigm is sensitive to both anxiolytic and anxiogenic agents [25,26]. The social interaction test (SIT) complements the EPM by capturing social aspects of anxiety, with reduced interaction between conspecifics indicating higher anxiety levels [27]. Using both tests in parallel allows complementary assessment of exploratory conflict and social behavior, supporting their selection for the present study.

Comment: Minor suggestions:

  • Adopt in vivo and in vitro in italics throughout.
  • Maintain consistent tense: use present tense for general knowledge and past tense for specific previous findings.
  • Change “facilitating the ‘fight or flight’ response” → “triggering the classical ‘fight-or-flight’ response” for fluency.
  • Replace “the extracts and combinations’ antioxidant activity” → “the antioxidant activity of the extracts and their combinations” for clarity.

Response:

This has been addressed in the revised manuscript.

Specific Comments

3.1. Plant Material

Comment: Please clarify the botanical identification procedure — indicate if plant material authentication was conducted by a botanist or relied solely on the supplier’s certificate

Response:

We relied solely on the accompanying certificate, as the plant material is offered chopped and packaged.

Comment: Replace “drug ratio EP to OA in a 1:1 (CE1) and EP to OA in a 3:1 (CE2)” →“drug-to-extract ratios of E. purpurea to O. acanthium were 1:1 (CE1) and 3:1 (CE2).”

Response: The correction has been made.

Comment: Consider merging the two first sentences for flow:

“Dry aerial parts of purpurea and flowers of O. acanthium (Asteraceae) were obtained from Herb Pharmacy 36.6 (Plovdiv, Bulgaria), accompanied by quality certificates from MediHerb-83 Ltd.”

Response: The correction has been made.

Comment: Typo: “deter-mined” → “determined.”

Response: The correction has been made.

3.3. Determination of Antioxidant Activity in vitro

3.3.1. ORAC assay

Comment: Replace “peroxid radicals” → “peroxyl radicals.”

Response: The correction has been made.

Comment: Express results uniformly: “Results were expressed as μmol Trolox equivalents per gram of extract (μmol TE/g).”

Response: The correction has been made.

Comment: Correct English syntax: “Measurements were performed using a FLUOstar OPTIMA fluorometer (BMG LABTECH, Offenburg, Germany) at excitation/emission wavelengths of 485/520 nm.”

Response: The correction has been made.

Comment: “Three independent replicates” → “All assays were conducted in triplicate.”

Response: The correction has been made.

3.3.2. HORAC assay

Comment: Correct “FLUOstarhu” → “FLUOstar.”

 Response: The correction has been made.

Comment: Replace “form complexes in conditions of Fenton reaction” → “assess complex formation under Fenton reaction conditions.”

Response: The correction has been made.

Comment: Ensure units consistency: “μmol gallic acid equivalents per gram of extract (μmol GAE/g).”

 Response: The correction has been made.

3.3.3. Electrochemical method

Comment: Correct typographical issues (“per litre/minute” → “per liter per minute”).

Response: The correction has been made.

 Comment: Clarify: “Antioxidant activity (AOA) was calculated using the kinetic criterion (K, μmol/L·min), comparing sample values to the Trolox standard according to the equation AOA = Ksample/KTrolox.”

Response: The correction has been made.

Comment: Prefer: “Each sample was tested in triplicate (n = 3).”

 Response: The correction has been made.

3.4.2. Experimental groups and treatments

Comment: Table legend should include the sample size (n = 8 per group).

 Response: The correction has been made.

 Comment: Clarify abbreviations at first use (EP, OA, CE1, CE2).

Response: The correction has been made.

Comment: Consider rewriting “to mimic the handling stress experienced by the treated groups” → “to control for handling-related stress.”

Response: The correction has been made.

Comment: Replace “distinct color (red)” → “a non-toxic red marker.”

Response: The correction has been made.

Comment: Correct “Reduced social engagement was interpreted as an indicator of social withdrawal and anxiety-like behavior” → “Reduced social engagement was interpreted as indicative of anxiety-like or socially avoidant behavior.”

Response: The correction has been made.

Comment: “The arena was cleaned with 70% ethanol…” → add “and dried before the next trial.”

Response: The correction has been made.

Minor suggestions

Comment: Use purpurea and O. acanthium in italics consistently.

Response: The correction has been made.

Comment: Replace “bw” → “body weight (bw)” on first appearance.

Response:  The correction has been made.

Comment: Keep all instrument names in parentheses.

Response: The correction has been made.

Comment: Double-check numbering — there is a duplicated “3.3.3” (electrochemical method appears twice). Renumber to maintain sequential order.

Response: We have double-checked the numbering of all subsections. We would like to clarify that the electrochemical method appears only once in the manuscript and the numbering follows the correct sequential order (3.3.1, 3.3.2, 3.3.3). No duplicated “3.3.3” was found. Nevertheless, we carefully re-verified the entire section to ensure full consistency in the structure and formatting.

Comment: Ensure consistent unit spacing: use “μmol/L·min,” “pg/mL,” and “°C” without extra spaces.

Response:  The correction has been made.

Results/Discussion

General comments

Comment: Overall structure: The section effectively integrates results with discussion, but could benefit from clearer segmentation between data presentation and interpretation. In several paragraphs, interpretation precedes quantitative data, which can confuse readers. Consider reorganizing to first present findings, followed by their implications.

Response:

The Discussion section has been reorganized as recommended.

Conciseness: Some paragraphs reiterate the same conclusion (e.g., the synergistic effect of CE2) multiple times. Streamlining these instances would improve readability and focus.

Interpretative depth: While biochemical rationale is strong (e.g., linking chlorogenic acid and IL-10), some claims could be further substantiated with quantitative or comparative language. For example: “CE2 demonstrated the highest activity and the most pronounced behavioral effects.” Could specify “approximately X-fold higher than EP or OA alone.”

Minor suggestions

Comment: Replace repetitive phrasing (“anxiolytic-like effects”) occasionally with synonyms such as “anxiety-reducing activity” or “anxiolytic potential” for stylistic variation.

Response: This has been addressed in the revised manuscript.

Use “respectively” sparingly — in several instances it can be removed for smoother readability.

Verify consistency of abbreviations (e.g., CE2 vs. C-E2, OAER vs. OAEr).

Response: The correction has been made.

Reviewer 2 Report

Comments and Suggestions for Authors

This is a very interesting work that compares two extracts and their combinations from the traditional medicinal plant Echinacea purpurea (L.) Moench and Onopordum acanthium (L., to study their antioxidant activity in vitro and Anxiolytic Effect in vivo. However, I have significant concerns regarding the analysis of the investigation. 

First of all, authors need to have an exhaustive edition. Tables and figures are very close to the paragraphs, and it isn’t very easy to understand some of them.

MAJOR CONCERNS

  1. My primary concern is that only one single dose was tested. What was the reason of this? And only one acute stress model was assayed, as well. Authors need to explain these situations. In my opinion, at least three doses need to be assayed.
  2. What was the reason to choose combinations of EP and OA in ratios of 1:1 and 3:1, respectively?
  3. The manuscript needs to be revised for grammatical mistakes.
  4. The conclusions section lacks conclusions; it is more akin to a discussion section. It needs to be rewritten.

Author Response

We thank the reviewer for the helpful and valuable comments and recommendations, which will help improve the quality of the manuscript. Below, we provide point-by-point responses to the comments. All changes are highlighted in red.

Comment: My primary concern is that only one single dose was tested. What was the reason of this? And only one acute stress model was assayed, as well. Authors need to explain these situations. In my opinion, at least three doses need to be assayed.

Response:

We appreciate the reviewer’s insightful comment regarding the use of a single dose and the employment of only one acute stress model. In this initial study, we selected a single dose of the extracts and their combinations (500 mg/kg bw) based on prior literature, with the primary aim of establishing proof-of-concept for their anxiolytic-like, immunomodulatory, and antioxidant effects. We agree that a dose–response analysis and the evaluation of additional stress models would provide a more comprehensive understanding of efficacy, minimum effective dose, and potential toxicity. However, due to resource limitations and ethical considerations regarding animal use, we restricted our investigation to one dose and one acute stress paradigm. We acknowledge this as an important limitation in the conclusion section.

Comment: What was the reason to choose combinations of EP and OA in ratios of 1:1 and 3:1, respectively?

Response: These ratios were selected based on our prior phytochemical and pharmacological data (Vlasheva et al., Plants 2024). We suggest combined extracts of E. purpurea and O. acanthium for the frst time. The ratio between the drugs 1:1 was chosen, similar to the combination of Echinacea and Hypericum perforatum studied by Bajrai et al. (Sci. Rep. 2022, 12, 21723), while the ratio 3:1 is entirely our proposal,following the idea of keeping the effect of E. purpurea but eliminating possible side effects

Comment: The manuscript needs to be revised for grammatical mistakes.

Response: The entire text has been thoroughly revised by a native English language editor (acknowledged in the manuscript) to correct grammar, syntax, and clarity.

Comment: The conclusions section lacks conclusions; it is more akin to a discussion section. It needs to be rewritten.

Response: The conclusion has been rewritten as recommended.

Reviewer 3 Report

Comments and Suggestions for Authors

1- The plagiarism of the research is 39% with 9% from one source (https://doi.org/10.3390/plants13233397 ).

  1. The title is long and requires rewriting (e.g., Antioxidant and Anxiolytic Effects of Echinacea purpurea and Onopordum acanthium Extracts and Their Combinations).
  2. The results of the abstract need to include more numerical data.
  3. The conclusion in the abstract should emphasize the most significant finding.
  4. Please clearly write the research gap in the introduction section.
  5. The extraction method requires more details, e.g., extraction yield, drying and storage conditions, and why 60% ethanol is used.
  6. Include ELISA kit catalog numbers and detection ranges.
  7. Please clarify whether samples were assayed in duplicate or triplicate in each experiment.
  8. We suggest including oxidative stress biomarkers (e.g., MDA, SOD, GPx) to link antioxidant effects with behavioral outcomes.
  9. In all results and discussion sections, add statistical comparisons between groups with p-values.
  10. It is good to compare the obtained ORAC/HORAC values with literature-reported values for similar extracts.
  11. It is good to include scatter plots for key correlations (IL-10 vs OATR, IFN-γ vs OAER).
  12. It is good to add a graphical abstract that summarizes the findings.
  13. The references contain many old references. Please update them with more recent literature.

Author Response

We thank the reviewer for the helpful and valuable comments and recommendations, which will help improve the quality of the manuscript. Below, we provide point-by-point responses to the comments. All changes are highlighted in red.

Comment: The plagiarism of the research is 39% with 9% from one source (https://doi.org/10.3390/plants13233397 ).

Response: The manuscript indeed shares methodological and background sections with our previous publication (Vlasheva et al., Plants 2024, 13, 3397), as this study is a continuation of that work using the same extracts and experimental approach. However, the current article presents novel in vivo data on the anxiolytic-like effects and cytokine modulation under acute cold stress — which were not included in the Plants paper. To minimize text overlap, we have paraphrased and shortened the sections describing materials and methods, ensuring that duplicated wording is reduced to the minimum necessary for reproducibility.

Comment: The title is long and requires rewriting (e.g., Antioxidant and Anxiolytic Effects of Echinacea purpurea and Onopordum acanthium Extracts and Their Combinations).

Response: We appreciate the reviewer’s suggestion. However, we prefer to keep the original title, as it accurately reflects the full scope of both the in vitro and in vivo components of the study. Shortening it would omit essential information and could misrepresent the breadth of the work.

Comment: The results of the abstract need to include more numerical data.

Response: The correction has been made.

Comment: The conclusion in the abstract should emphasize the most significant finding.

Response: The conclusion in the abstract has been revised to emphasize the main findings -namely, the attenuation of anxiety-like behavior and the modulation of the immune response through up-regulation of IL-10 observed with OA and CE2.

Comment: Please clearly write the research gap in the introduction section.

Response: This has been addressed in the revised manuscript.

Comment: The extraction method requires more details, e.g., extraction yield, drying and storage conditions, and why 60% ethanol is used.

Response:  To calculate the yield, the concentrated aqueous extract (described in section 3.2.) was further dried entirely for one day. The yield obtained was 37.5%. The extractions were performed with 60% ethanol since it was found that waterethanol mixtures optimally extract the phenolic compounds.

Comment: Include ELISA kit catalog numbers and detection ranges.

Response: ELISA kit catalog numbers and detection ranges have been added in Material and Methods section, subsection 3.4.4.3. Immunological assays.

Comment: Please clarify whether samples were assayed in duplicate or triplicate in each experiment.

Response: Antioxidant activity was determined in triplicates to ensure higher precision, and immunological assays were performed in duplicates according to the manufacturer’s recommendations. This is clearly outlined in the relevant sections of the Materials and Methods.

Comment: We suggest including oxidative stress biomarkers (e.g., MDA, SOD, GPx) to link antioxidant effects with behavioral outcomes.

Response:Thank you for the suggestion, but this was not the subject of this study.

Comment: In all results and discussion sections, add statistical comparisons between groups with p-values.

Response: This has been addressed in the revised manuscript.

Comment: It is good to compare the obtained ORAC/HORAC values with literature-reported values for similar extracts.

Response:There is not much information about such extracts in the available literature. We have used data from the work of Fu et al., 2021 and Parzhanova et al., 2023 for comparison.

Comment: It is good to include scatter plots for key correlations (IL-10 vs OATR, IFN-γ vs OAER).

Response: Scatter plot for the correlation that has been found to be significant (IL-10 vs OATR) has been added (Fig. 6).

Comment: It is good to add a graphical abstract that summarizes the findings.

Response: It has been added.

Comment: The references contain many old references. Please update them with more recent literature.

Response: Several new references from the last years have been added (References № 9, 19, 26, 54). Although some of the cited articles are from earlier periods, they are relevant because they describe the established behavioral methods and techniques for assessing anxiolytic-like effects and antioxidant activity, respectively, which were employed in the present study.

Reviewer 4 Report

Comments and Suggestions for Authors

Dear authors,

The manuscript entitled “Comparative Study on Extracts from Traditional Medicinal Plants Echinacea purpurea (L.) Moench and Onopordum acanthium (L.): Antioxidant Activity in vitro and Anxiolytic Effect in vivo”deals with evaluation of extracts as we as extracts combinatons from from Traditional Medicinal Plants Echinacea purpurea (L.) Moench and Onopordum acanthium (L.) for their antioxidant and anxiolytic activity.The manuscript is well documented and well written. A little bit more details should be given in experimental parts of 3.3.1 and 3.3.2. parts.

Page 2. Line 88. Should be: Contrary wise,

Page 14. Line 447.Should be: determined

Page 14.Line 466.Shoiuld be: H2O2

Page 14. Line 467. Should be: extract

Page 15. Line 498. Should be : E. purpurea, O. acanthium

Page 15. Line 499.Should be: combination

Page 15. Line 500. Should be: combination

Comments on the Quality of English Language

Quality of English language is more or less good.

Author Response

We thank the reviewer for the helpful and valuable comments and recommendations, which will help improve the quality of the manuscript. Below, we provide point-by-point responses to the comments. All changes are highlighted in red.

Comment: A little bit more details should be given in experimental parts of 3.3.1 and 3.3.2. parts.

Response: They have been added.

Comment: Page 2. Line 88. Should be: Contrary wise,

Response: Corrected

Comment: Page 14. Line 447.Should be: determined

Response: We would like to clarify that in this construction (“was used to…”), the verb following to remains in the infinitive form. Therefore, the correct phrasing is “was used to determine the AOA.” We have verified the sentence and confirm that it is grammatically correct in its current form.

Comment: Page 14.Line 466.Shoiuld be: H2O2

Response: The correction has been made.

Comment: Page 14. Line 467. Should be: extract

Response: The correction has been made.

Comment: Page 15. Line 498. Should be : E. purpurea, O. Acanthium

Response: The correction has been made.

Comment: Page 15. Line 499.Should be: combination

Response: The correction has been made.

Comment: Page 15. Line 500. Should be: combination.

Response: The correction has been made.

Round 2

Reviewer 1 Report

Comments and Suggestions for Authors

Dear authors,

I would like to congratulate you for the efforts to improve the manuscript.

All my concerns were properly addressed.

Reviewer 2 Report

Comments and Suggestions for Authors

I agree with the changes in the manuscript

Reviewer 3 Report

Comments and Suggestions for Authors

Authors addressed most of the raised comments.